# *Chiliadenus iphionoides*: From chemical profiling to anticancer, antioxidant, *α*-amylase, and *α*-glycosidase activities

Odey Bsharat[1]*, Yousef Salama[2], Nisreen Al-Hajj[1], Nawaf Al-Maharik[1]*

1 Department of Chemistry, Faculty of Sciences, An-Najah National University, Nablus, Palestine,
2 An-Najah Center for Cancer and Stem Cell Research, Faculty of Medicine and Health Sciences, An-Najah National University, Nablus, Palestine

* obsharat@najah.edu (OB); n.maharik@najah.edu (NA-M)

## Abstract

Recently, essential oils (EOs) have garnered attention for their biological properties as a source of natural compounds with anticancer, antibacterial, and antioxidant effects. *Chiliadenus iphionoides* (Boiss.& Blanche) Brullo is a fragrant aromatic species indigenous to Palestine and the neighboring countries. It is utilized by the Bedouins for the treatment of many ailments. This study aimed to analyze the chemical composition of essential oil extracted from the desiccated leaves of *Chiliadenus iphionoides* (Boiss.& Blanche) Brullo and to assess its in vitro antioxidant, anticancer, and α-amylase and lipase inhibitory activities. The EO obtained during hydrodistillation was analyzed using gas chromatography-mass spectrometry to ascertain their chemical composition. GC-MS analysis of the EO from dried *Chiliadenus iphionoides* (Boiss.& Blanche) Brullo leaves identified 47 compounds, comprising 98.81% of the total oil. The predominant constituents included cresol methyl ether (52.93%), ethyl 2-octynoate (14.36%), epi-α-cadinol (6.56%), 1,8-cineole (4.25%), and 7-epi-α-eudesmol (3.66%). The EO exhibited significant antioxidant activity against 1,1-diphenyl-2-picrylhydrazyl (DPPH) with an $IC_{50}$ value of 19.83±0.99 µg/mL. Nevertheless, it exhibited limited lipase activity and subpar α-amylase activity. *Chiliadenus iphionoides* (Boiss.& Blanche) Brullo EO exhibited significant cytotoxicity against B16F10 and MCF-7 cancer cell lines, with $IC_{50}$ values of 8.74±0.11 and 13.68±0.17 µg/mL, respectively. The combination of the anticancer medication Taxol (10 ng) with *Chiliadenus iphionoides* (Boiss.& Blanche) Brullo EO at concentrations of 25 µg/mL and 50 µg/mL significantly enhanced the inhibitory efficacy of Taxol against MCF-7 and B16F10 cells.

**Data availability statement:** • Availability of data and materials: The datasets generated and/or analysed during the current study are available in the study.

**Funding:** The author(s) received no specific funding for this work.

**Competing interests:** NO authors have competing interests.

## 1. Introduction

Herbal medicine, commonly referred to as phytomedicine, has gained widespread recognition worldwide. Recent statistics reveal that more than 80% of individuals in developing countries rely on herbal medicine, while there is a growing interest in complementary and alternative medicine among those in wealthier nations [1,2]. Furthermore, around 50% of the pharmaceuticals utilized in contemporary medicine derived or inspired from medicinal plants [3]. Numerous studies have demonstrated that plants contain a wealth of bioactive phytochemicals [4,5], particularly essential oils (EOs), which are vital in the prevention of chronic diseases, including infectious, cancer, diabetes and vascular [6]. EOs exhibit diverse pharmacological and biological activities, encompassing anti-inflammatory, antibacterial, antioxidant, anticancer, and antidiabetic actions [7,8]. They are utilized to combat germs responsible for serious infectious disorders owing to their antibacterial, antifungal, and antiviral properties [9]. Consequently, the empirical data endorsing these activities may validate the conventional uses associated with these herbs. There has been considerable scientific interest in investigating the bioactivities of EOs and their bioactive constituents. The anticancer characteristics of natural products, particularly EOs, are pivotal in the development of novel pharmaceuticals to combat the second leading cause of mortality globally, while minimizing adverse effects [10]. Moreover, assessing the antioxidant capacity of plants can reveal their potential to neutralize harmful free radicals, hence mitigating oxidative stress, which is commonly associated with the onset of diabetes [11]. The increasing prevalence of diabetes globally compels scientists to seek viable options for its management. Consequently, assessing the biological properties of medicinal plant extracts may result in the identification of novel, safe, and effective bioactive molecules exhibiting antibacterial, antioxidant, and antidiabetic activities.

In this context, we focused on the potential medicinal plant *Chiliadenus iphionoides* (Boiss.& Blanche) Brullo due to its application in traditional medicine in Palestine and adjacent regions as a decoction or infusion for treating influenza, colds, fever, abdominal pain, depression, anxiety, ocular infections, and nephrolithiasis; it has also been utilized as an antispasmodic [12]. *Chiliadenus iphionoides* (Boiss.& Blanche) Brullo is one of ten species in the small genus *Chiliadenus* in the *Asteraceae* family. This tiny, shrubby perennial possesses aromatic leaves, hairy and adhesive stems, and tubular yellow flowers that bloom from September to December, usually found in Palestine, Syria, Lebanon, and Jordan. The little leaves are adorned with trichomes and glands that secrete essential oil, imparting the plant its distinctive fragrance. Notwithstanding the extensive application of the plant in traditional Middle Eastern medicine, only a few investigations has been conducted to assess the chemical composition of the essential oil of *Chiliadenus iphinoides* collected in Jordan.

To the best of our knowledge, the chemical profile and biological properties of *Chiliadenus iphinoides* found in Palestine have not been previously examined. Considering our focus on the phytochemical composition of essential oils from aromatic plants native to Palestine, this study investigated the phytochemical profile of EOs extracted

from both dry leaves of Chiliadenus iphinoides, gathered from Jericho, employing GC/MS analysis. Furthermore, we examined their activities related to antioxidant properties, anticancer effects, α-amylase inhibition, and lipase inhibition.

## 2. Materials and methods

### 2.1. Species collection and identification

The plant material of *Chiliadenus iphionoides* (Boiss.& Blanche) Brullo utilized in this study was collected from Jericho, Palestine, in early September 2023 [geographical coordinates: latitude 31° 52' 0 N, longitude 35° 27' 0 E, altitude 276 m below sea level]. Pharmacologist Dr. Nidal Jaradat identified the plant, and the voucher specimens, labelled Pharm-PCT-2814, were stored at the Herbal Products Laboratory at An-Najah National University.

### 2.2. Extraction of essential oils

*Chiliadenus iphionoides* (Boiss.& Blanche) Brullo leaves were dried in a shaded location at room temperature ($25 \pm 3°C$) and humidity ($55 \pm 4$ RH), then crushed into small pieces and stored for later use. The EO was extracted for three hours using hydrodistillation with 100 grams of dried, crushed leaves and 300 mL of distilled water in a 500-milliliter round-bottom flask attached to a Clevenger apparatus. The EO was extracted using diethyl ether ($Et_2O$, 10 mL x 2), the combined organic phases were dried over calcium chloride, and the solvent was evaporated under reduced pressure, resulting in a yield of 0.7% pale-yellow oil. The EO was kept in clean glass vials with a secure lid until needed.

### 2.3. Qualitative and quantitative analysis of the extracted EOs

Gas chromatography-mass spectrometry was employed to analyze the chemical composition of the EO from *Chiliadenus iphionoides* (Boiss.& Blanche) Brullo both qualitatively and quantitatively. This task was achieved using an HP 5890 Series II gas chromatograph fitted with a nonpolar Perkin Elmer-5-MS capillary column, which has an inner diameter of 0.25 mm, a length of 30 m, and a film thickness of 0.25 μm. 1 μL of essential oil, produced at a concentration of 1000 parts per million (ppm), was injected. Helium served as the carrier gas at a flow rate of 1 mL/min, maintained under a pressure of 20.41 psi in split mode, with a split ratio of 1:50. The injector and the transfer line were both maintained at a temperature of 250°C. The oven was first set to 50°C for 5 minutes, subsequently increasing to 280°C at a rate of 4°C per minute. The sample was maintained at a constant temperature of 280°C for 10 minutes. The run lasted 62.5 minutes, followed by a 10-minute conditioning period. The detection was conducted using a Perkin Elmer Clarus 560 mass spectrometer. The acquisition utilized electron ionization (EI) mode with an ionization voltage of 70 eV, operating in standard scanning mode across a mass range of 40–500 m/z. Individual metabolites were identified by comparing their relative retention indices (RRI) and mass spectral data with the MS library, NIST webbook, and pertinent literature, in addition to analyzing their fragmentation patterns. The detected compounds were expressed as percentages of the peak area of each individual component relative to the total peak area of the EO. The RRI for each phytoconstituent was established by comparing their retention times with those of a standard solution of n-alkanes (C7–C30) under identical experimental conditions.

### 2.4. DPPH Free radical scavenging assay

The DPPH radical method was used to assess the antioxidant activity of the EO extracted from *Chiliadenus iphionoides* (Boiss.& Blanche) Brullo following a reported literature procedure [13,14]. The preliminary phase involved the formulation of a 0.1 mM DPPH solution in methanol. A stock solution of EO in methanol at a concentration of 100 μg/mL was prepared, followed by the preparation of a Trolox solution as the positive control. Six dilutions of both EO and Trolox—5, 10, 20, 50, 80, and 100 μg/ml—were produced. One milliliter of the methanolic DPPH solution was incorporated into each essential oil dilution. Upon the addition of 1 mL of methanol, the ultimate working capacity reached 3 mL. The blank control was established by dissolving DPPH in methanol at a 1:2 ratio without the inclusion of EO. Subsequently, all solutions were

allowed to remain at room temperature in total darkness for thirty minutes. The absorbance of the solutions was measured using a UV-Vis spectrophotometer calibrated to 517 nm following the incubation period. The DPPH radical scavenging capacity was calculated using the following formula: Scavenging capacity $= A_{control} - A_{sample} / A_{control} \times 100$ (1), where $A_{control}$ is the absorbance of DPPH radical without any additive and $As_{ample}$ is the absorbance of DPPH radical with oil samples and control solutions of various concentrations. A graph of scavenging activity as a percentage inhibition was plotted against the concentrations of essential oils and standards to determine their $IC_{50}$ value.

## 2.5. *α*-Amylase inhibition assay

The α-Amylase activity of the EO was determined by the Worthington enzyme method [12]. The EO (100 mg) was first dissolved in a small amount of 10% DMSO, followed by the addition of 0.02 M $Na_2HPO_4/NaH_2PO_4$ and 0.006 M NaCl buffer solution (pH 6.9) to a final volume of 100 mL, creating a stock solution with a concentration of 1 mg/mL. From this stock solution, five concentrations of the EO were prepared for bioassay by dissolving the EO in 10% DMSO to prepare the following dilutions, respectively: 10, 50, 70, 100, 500, and 1000 µg/mL. A 0.2 mL aliquot of EO solution was mixed with 0.2 mL of porcine pancreatic α-amylase (2 units/mL) and incubated at 30°C for 10 min. After a 10-minute incubation at 30°C, 0.2 mL of freshly prepared 1% starch solution was added, and the mixture was incubated for an additional 3 minutes. The reaction was terminated by adding 3,5-dinitrosalicylic acid (DNSA), followed by dilution with 5 mL of distilled water. The mixture was then heated at 90°C in a water bath for 10 minutes. After cooling to room temperature, the absorbance was measured at 540 nm. The α-amylase inhibitory activity was calculated according to the equation: α-amylase inhibition (%) = $(A_B-A_E)/A_B \times 100$, where $A_B$ is the absorbance of the blank solution and $A_E$ is the absorbance of the EO. A graph of *α*-amylase inhibition (%) was plotted against the concentrations of oils and standards to determine their $IC_{50}$ value.

## 2.6. Porcine pancreatic lipase inhibition assay

The inhibitory activities of the EO were examined following a previously established procedure [15]. A stock solution of EO was prepared at a concentration of 500 µg/mL in 10% DMSO. A dilution series was prepared from the stock solution, resulting in five distinct concentrations of 50, 100, 200, 300, and 500 µg/mL. A stock solution of porcine pancreatic lipase (1 mg/mL) was prepared in Tris-HCl buffer at pH 8.5. A stock solution of p-nitrophenyl butyrate (PNPB) was prepared by dissolving 20.9 mg in 2 mL of acetonitrile. Test samples (100 µL) and pancreatic lipase solution (200 µL, 0.8 µg/mL) were prepared in Tris-HCl buffer to reach a total volume of 1 mL. The mixture was kept in darkness at 37°C for 15 minutes. Following this, 20 µL of p-nitrophenyl palmitate (4 µg/mL) was introduced and incubated for 30 minutes at 37°C. Analyses were performed at a wavelength of 450 nm. The α-lipase inhibitory activity was determined using the equation: lipase inhibition (%) = (AB-AE)/AB × 100, where AB is the absorbance of the blank solution and AE is the absorbance of the EO. A graph illustrating lipase inhibition was created to determine the $IC_{50}$ value in relation to the concentrations of oils and standards.

## 2.7. Cytotoxicity of EO

### 2.7.1. Cell lines and cell culture.
The B16F10 melanoma cell lines (ATCC CRL-6475) were cultured in Dulbecco's Modified Eagle Medium (DMEM, Gibco, Waltham, Massachusetts, USA) enriched with high glucose, L-glutamine, phenol red (Wako, Japan), 10% fetal bovine serum (FBS; HyClone, Logan, UT, USA), and 1% penicillin/streptomycin (P/S) (Sigma, USA). MCF-7 breast cell lines (ATCC HTB-22) were cultured in ATCC-formulated Eagle's Minimum Essential Medium, Catalog No. 30–003. The basal medium was supplemented with 0.01 mg/mL of human recombinant insulin, fetal bovine serum to a final concentration of 10%, and 1% penicillin-streptomycin. The cells were incubated at 37°C in a 5% $CO_2$ atmosphere.

### 2.7.2. Chemicals and EOs.
The EOs stock solutions were prepared by dissolving them in DMSO to a concentration of 500 µg/mL.

**2.7.3. Cell Cultures with EOs.** B16F10 and MCF-7 cells ($2 \times 10^5$ cells/well) were cultured in 6-well plates (TPP, Switzerland) and incubated overnight prior to the introduction of DMSO (EO control), EO, and Mix at doses between 5 and 100 µg/mL EO. At 24 hours, viable cells were enumerated utilizing trypan blue dead-cell exclusion dye (Sigma, USA). In several tests, B16F10 or MCF-7 cells were grown with or without Taxol (10 ng/mL) with essential oils [16].

## 2.8. Statistical analysis

All experiments were performed at least three times. Data are shown as the mean +/- standard error of the mean (SEM). Student's t-test or ANOVA with Tukey HSD post hoc tests using the R program were used. Prism software used to determine $IC_{50}$.

# 3. Results and discussion

## 3.1. Phytochemistry

*Chiliadenus iphionoides* (Boiss.& Blanche) Brullo EO was obtained via hydrodistillation as a pale-yellow oil in 0.7% yield. The GC-MS analysis of the EO identified 47 components, which constitute 98.81% of the total oil, as illustrated in Table 1 and Fig 1. The main components are cresol methyl ether (52.93%), ethyl 2-octynoate (14.36%), epi-α-cadinol (6.56%), 1,8-cineole (4.25%), and 7-epi-α-eudesmol (3.66%). The oxygenated monoterpenes and oxygenated sesquiterpenes are present in the EO with notable values of 11.0% and 9.77%, respectively. While sesquiterpene hydrocarbons and monoterpene hydrocarbons are less abundant, with values of 2.37% and 0.77%, respectively. The chemical constituents of the essential oil (EO) extracted from the leaves of Chiliadenus iphionoides (Boiss. & Blanche) Brullo, collected from the lowest point on Earth (Jericho), differ among the five primary chemotypes obtained from various locations, as documented in the literature; nonetheless, there is a general agreement concerning the predominant class. Avato et al. reported the identification of forty-five compounds in the EO from *Chiliadenus iphionoides* (Boiss.& Blanche) Brullo aerial parts growing in Jordan, accounting for 90.2% of the oil, with monoterpenes being the most abundant group [17]. The study revealed the presence of borneol as the major constituent (49.3%), followed by 1,8-cineole (8.4%), α-terpineol (3.8%), camphor (3.7%), bornyl formate (3.6%), terpin-4-ol (3.0%), bornyl acetate (2.9%), and selin-11-en-4-α-ol (2.4%) [17]. AlNaimat et al. reported the identification of 23 compounds in EOs from the aerial parts of three *Chiliadenus iphionoides* (Boiss.& Blanche) Brullo species at full flowering stage, of which eucalyptol (42.6%), trans-chrysanthemol (20.65%), Yomogi alcohol (10.04%), γ-terpinene (4.09%), and o-cymene (3.40%) were the major components [18]. A study at Hebrew University demonstrated considerable variability in essential oil samples from wild plants across various regions of historical Palestine, establishing a clear correlation between their chemical composition and geographical location, identifying three primary chemotypes: camphor/a-pinene/pochienol, t-cadinol/1,8-cineole/trans-chrysanthemol, and intermedeol [19].

## 3.2. Evaluation of the antioxidant

The antioxidant capacity of *Chiliadenus iphionoides* (Boiss.& Blanche) Brullo EO was evaluated through DPPH free radical scavenging assays, with results presented as an $IC_{50}$ value [20]. According to Fig 2, the extract of *Chiliadenus iphionoides* (Boiss.& Blanche) Brullo has a notable antioxidant capacity, evidenced by an $IC_{50}$ value of 19.83 ± 0.99 µg/mL, which is less effective than the Trolox (positive control) with an $IC_{50}$ value of 1.94 ± 0.10 µg/mL. The high concentrations of cresol methyl ether and oxygenated monoterpenes (11.0%) and oxygenated sesquiterpenes (9.77%) may contribute to the significant antioxidant activity observed in the EO. Furthermore, the presence of 1,8-cineole at a concentration of 4.25% may have significant implications, as numerous studies have demonstrated that 1,8-cineole exhibits a diverse array of pharmacological effects, primarily functioning as a strong antioxidant [21]. AlNaimat et al. conducted a recent assessment of the antioxidant capacity of EO derived from *Chiliadenus iphionoides* (Boiss.& Blanche) Brullo in Jordan, employing the DPPH method. This investigation produced an $IC_{50}$ value of 1.29 ± 0.3 mg TE/g DW, indicating significant antioxidant properties

**Table 1. Phytochemical composition of dry *Chiliadenus iphionoides (Boiss.& Blanche) Brullo* leaves.**

| No. | Compound | RT | RI | % Content |
|---|---|---|---|---|
| 1 | Cresol methyl ether | 12.48 | 990 | 52.93 |
| 2 | Anisole | 13.53 | 1014 | 0.36 |
| 3 | p-cymene | 13.85 | 1022 | 0.08 |
| 4 | 1,8-Cineole | 14.08 | 1027 | 4.25 |
| 5 | Benzene acetaldehyde | 14.67 | 1041 | 0.05 |
| 6 | γ -Terpinene | 15.31 | 1056 | 0.60 |
| 7 | cis-Vertocitral C | 15.98 | 1072 | 0.12 |
| 8 | p-Mentha-2,4(8)-diene | 16.45 | 1083 | 0.06 |
| 9 | Pinene oxide | 17.18 | 1100 | 0.30 |
| 10 | 1,3,8-p-Menthatriene | 17.43 | 1107 | 0.09 |
| 11 | 1,5,7-Octatrien-3-ol, 3,7-dimethyl- | 17.54 | 1109 | 0.02 |
| 12 | trans-a-Necrodol | 18.61 | 1137 | 0.13 |
| 13 | Camphor | 18.87 | 1144 | 0.14 |
| 14 | Nerol oxide | 19.1 | 1149 | 0.75 |
| 15 | Borneol | 19.93 | 1171 | 1.72 |
| 16 | Terpinen-4-ol | 20.23 | 1193 | 1.49 |
| 17 | Methyl 2-octynoate | 21.12 | 1201 | 0.13 |
| 18 | trans-Dihydrocarvone | 21.22 | 1204 | 1.25 |
| 19 | Pulegone | 22.34 | 1235 | 1.66 |
| 20 | Ethyl 2-octynoate | 23.73 | 1274 | 14.36 |
| 21 | lavandulyl acetate | 23.96 | 1281 | 0.75 |
| 22 | Neryl acetate | 26.49 | 1355 | 0.41 |
| 23 | Ethyl cis-cinnamate | 27.03 | 1371 | 0.03 |
| 24 | Carvyl acetate | 27.33 | 1380 | 0.14 |
| 25 | Z-Jasmone | 27.66 | 1390 | 0.19 |
| 26 | β-Caryohyllene | 28.56 | 1418 | 0.26 |
| 27 | α-Trans-Bergamotene | 28.96 | 1431 | 0.09 |
| 28 | Aromadendrene | 29.38 | 1441 | 0.02 |
| 29 | α–Humulene | 29.69 | 1455 | 0.02 |
| 30 | allo-aromadendrene | 29.85 | 1457 | 1.23 |
| 31 | Ethyl E-cinnamate | 30 | 1464 | 0.05 |
| 32 | Nd | 30.09 | | 0.03 |
| 33 | Isomethyl- α-ionone | 30.25 | 1472 | 0.48 |
| 34 | Germacrene D | 30.49 | 1480 | 0.28 |
| 35 | 10,11-Epoxycalamenene | 30.68 | 1486 | 0.14 |
| 36 | β-Selinene | 30.95 | 1495 | 0.06 |
| 37 | α –Muurolene | 31.04 | 1498 | 0.03 |
| 38 | Lavandulyl isovalerate | 31.28 | 1505 | 0.03 |
| 39 | γ -Cadinene | 31.48 | 1512 | 0.19 |
| 40 | δ--Cadinene | 31.64 | 1518 | 0.19 |
| 41 | Nd | 32.14 | 1535 | 0.11 |
| 42 | 4-[(2E)-2-Butenyl]-1,2-dimethylbenzene | 32.55 | 1548 | 0.31 |
| 43 | Nd | 33.14 | 1568 | 0.06 |
| 44 | Nd | 33.8 | 1591 | 0.21 |
| 45 | Nd | 33.99 | | 0.24 |

*(Continued)*

**Table 1.** (Continued)

| No. | Compound | RT | RI | % Content |
|---|---|---|---|---|
| 46 | epi-α-Cadinol | 35.37 | 1647 | 6.56 |
| 47 | α-Eudesmol | 35.72 | 1659 | 2.74 |
| 48 | 7-epi-a-Eudesmol | 36.11 | 1672 | 3.66 |
| 49 | 5-neo-Cedranol | 36.63 | 1691 | 0.33 |
| 50 | Nd | 40.56 | 1842 | 0.30 |
| 51 | Nd | 40.72 | 1848 | 010 |
| | **Total identified%** | | | 98.81 |
| | **Monoterpene hydrocarbons** | | | 0.77 |
| | **Oxygenated monoterpenes** | | | 11.05 |
| | **Sesquiterpene hydrocarbons** | | | 2.37 |
| | **Oxygenated sesquiterpenes** | | | 9.77 |
| | **Others** | | | 74.85 |

**RT: Retention time, RI: Retention index.**

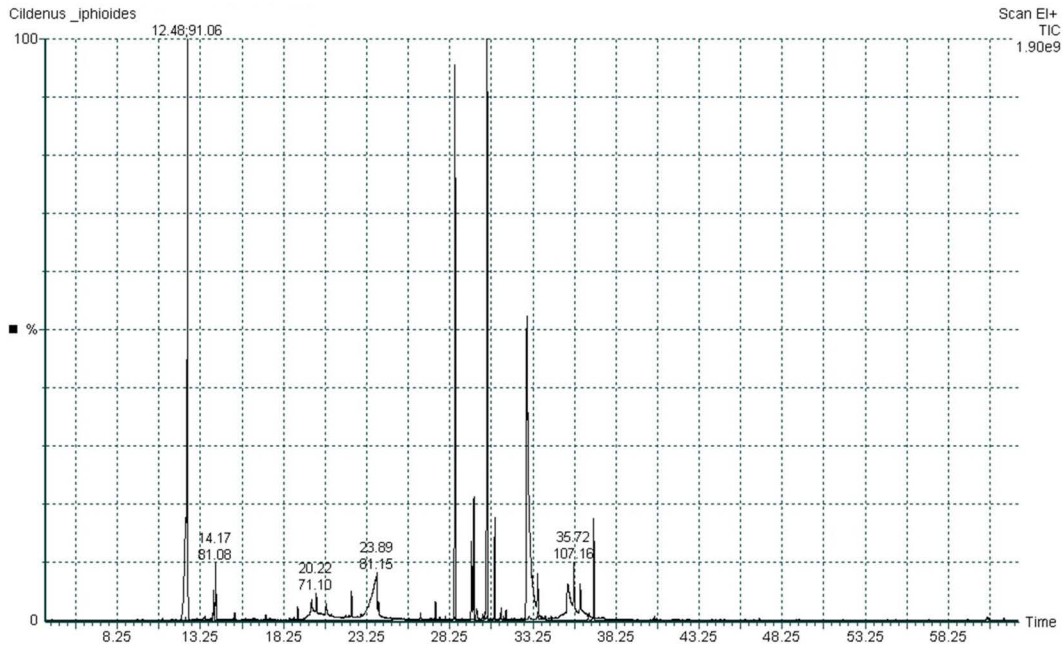

**Fig 1. GC chromatogram of *Chiliadenus iphionoides* (Boiss.& Blanche) Brullo EO.**

[18]. Al-Dabbas et al. discovered that the ethanol and water extracts of *Chiliadenus iphionoides* (Boiss.& Blanche) Brullo exhibited a markedly superior degree of DPPH radical-scavenging activity compared to butylated hydroxytoluene (BHT), achieving approximately 90% inhibition at a concentration of 100 mg/mL, attributed to the substantial presence of poly-phenols [22]. Sbieh et al. reported that the aqueous, methanol, acetone, and hexane extracts of *Chiliadenus iphionoides* (Boiss.& Blanche) Brullo obtained from Palestine exhibited significant DPPH radical-scavenging activities, with $IC_{50}$ values measured at 5.2, 7.5, 4.2, and 218 µg/mL, respectively [23].

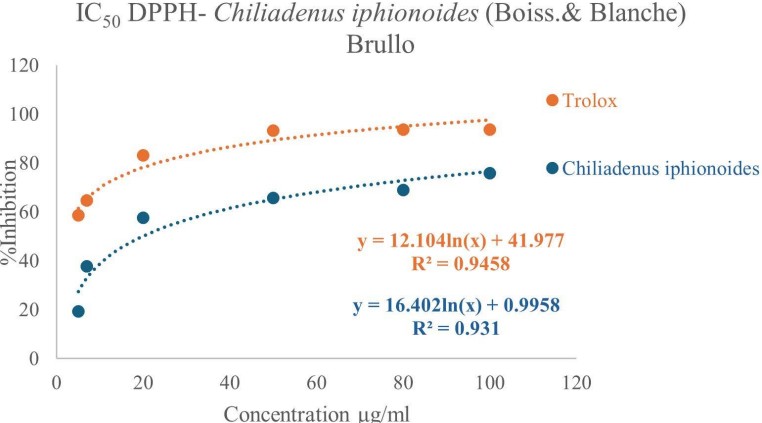

IC$_{50}$ DPPH- *Chiliadenus iphionoides* (Boiss.& Blanche) Brullo

**Fig 2. A plot of percent inhibition against the concentrations of *Chiliadenus iphionoides (Boiss.& Blanche) Brullo* EO extracted from dry leaves and Trolox as a positive control.**

### 3.3. Cytotoxicity of the *Chiliadenus iphionoides* EO

Cancer is one of the leading causes of death worldwide. In light of the considerable adverse effects linked to existing chemotherapeutic therapies [24], researchers are exploring a new candidate that demonstrates less toxicity and improved efficacy against cancer. The efficacy of *Chiliadenus iphionoides* (Boiss.& Blanche) Brullo in cancer treatment has not been comprehensively investigated, but it has historically been employed for other conditions. The cytotoxicity of *Chiliadenus iphionoides* (Boiss.& Blanche) Brullo EO was evaluated against melanoma (B16F10) and breast cancer (MCF-7) cell lines, as seen in Figs. 3 and 4. The examined EO demonstrated significant cytotoxicity against B16F10 and MCF-7 cell lines, revealing a dose-dependent decrease in cell viability, with IC$_{50}$ values of $5.70 \pm 0.11$ µg/mL and $13.68 \pm 0.17$ µg/mL, respectively as shown in Table 2.

A combination of equal parts of *Chiliadenus iphionoides* (Boiss.& Blanche) Brullo EO and *Teucrium polium L.* EO, characterized by E-nerolidol, geranyl acetone, germacrene D, and b-caryophyllene as predominant constituents, enhanced the potency of both oils against MCF-7 cancer cells, yielding an IC$_{50}$ value of $4.691 \pm 0.15$ µg/mL; however, there was only a

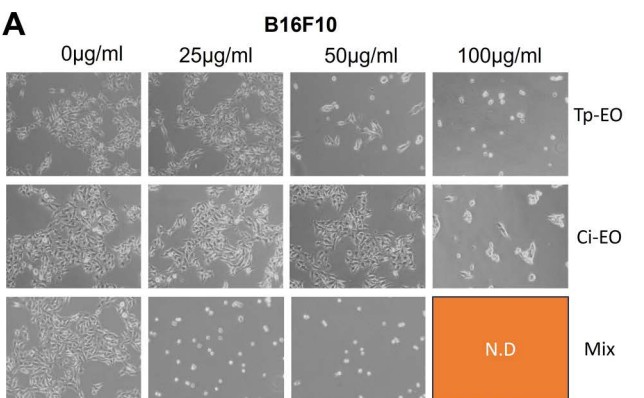
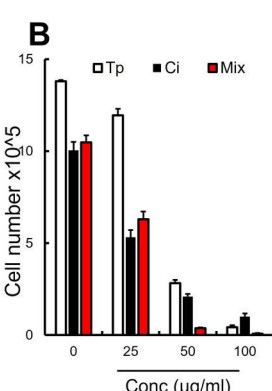

**Fig 3. (A) Macroscopic images of B16F10 cells after 24 hours of *T. polium* (Tp-EO), *C. iphionoides* (Ci-EO) and Mix treatment or DMSO as a control (n = 3 groups).** The indicated doses of EO's were applied to B16F10 cells in (B). After 24 hours treatment (n = 6). Viable cells were counted using Trypan blue after 24 h (n = 6/group). Data are expressed as mean +/- SEM. **p 0.01; *p 0.05 (Student's t-test).

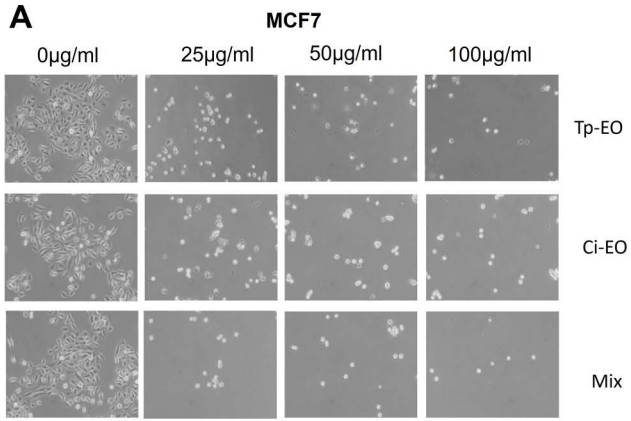
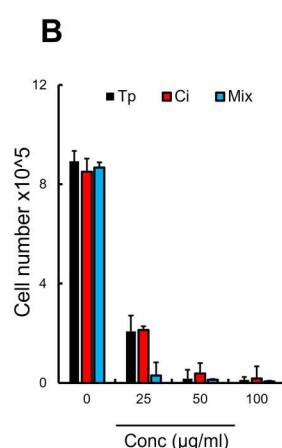

**Fig 4. (A) Macroscopic images of MCF-7 cells after 24 hours of *T. polium* (Tp-EO), C. iphionoides Ci-EO and Mix treatment or DMSO as a control (n = 3 groups).** The indicated doses of EO's were applied to MCF-7 cells in (B). After 24 hours treatment (n = 6). Viable cells were counted using Trypan blue after 24 h (n = 6/group). Data are expressed as mean +/- SEM. **p 0.01; *p 0.05 (Student's t-test).

**Table 2.** $IC_{50}$ values (µg/ml) for cytotoxicity of *Chiliadenus iphionoides* (Boiss.& Blanche) Brullo, combination of *Chiliadenus iphionoides* (Boiss.& Blanche) Brullo essential oil and *Teucrium polium L.* essential oil, combination of *Chiliadenus iphionoides* (Boiss.& Blanche) Brullo essential oil and 10 ng of taxol against MCF-7 and B16F10 cells.

| Name of Sample | ($IC_{50}$ value µg/mL) MCF-7 | $IC_{50}$ value µg/mL B16F10 |
|---|---|---|
| *Chiliadenus iphionoides* | 13.68 ± 0.17 | 8.74 ± 0.11 |
| *Chiliadenus iphionoides* + *Teucrium polium L* | 4.691 ± 0.15 | 5.984 ± 0.12 |
| *Chiliadenus iphionoides* + 10 ng of taxol | 2.788 ± 0.04 | 2.613 ± 0.05 |

marginal alteration in efficacy against B16F10 cells, as indicated in Table 2. The addition of 10 ng/mL of Taxol to Chiliadenus iphionoides essential oil at concentrations of 5 µg/mL, 25 µγ/mL, and 50 µg/mL markedly improved the inhibition of MCF-7 and B16F10 cancer cells, lowering the $IC_{50}$ to 2.788 and 2.613 µg/mL, respectively (Table 2). After 24 hours, Taxol (10 ng/mL) reduced cancer cells by 75.42%, whereas the combination of 10 ng of Taxol with 25 µg/mL and 50 µg/mL of EO led to a significant inhibition of MCF-7 cancer cells by 91.96% and 99.5%, respectively. The combination of Taxol (10 ng) with 25 µg/mL and 50 µg/mL of EO markedly improved the efficacy of Taxol, achieving an inhibition of B16F10 cancer cells of 89.45% and 93.95%, respectively, as seen in Fig 5.

AlNaimat et al. recently demonstrated that *Chiliadenus iphionoides* (Boiss.& Blanche) Brullo EO exhibited a dose-dependent inhibition of cell proliferation in A549 (human lung cancer adenocarcinoma), MDA-MB-231 (triple-negative breast cancer), T47 (human breast cancer), and EMT6/P (mouse mammary sarcoma) cell lines, with $IC_{50}$ values of 1.18 ± 0.04, 1.51 ± 0.04, 0.08 ± 0.009, and 0.03 ± 0.03 mg/mL, respectively. The significant anticancer activity observed in T47, and EMT6/P was partially attributed to 1,8-cineole [18].

### 3.4. Evaluation of the anti-lipase, and anti-*α*-amylase activities

The inhibitory effects of EO on lipase and α-amylase were assessed. The EO extracted from *Chiliadenus iphionoides* (Boiss.& Blanche) Brullo, as shown in Fig 6, demonstrated modest lipase inhibition activity, with $IC_{50}$ = 931 ± 46.55 µg/mL. To the best of our knowledge, this is the first reported lipase inhibition activity for the *Chiliadenus iphionoides* (Boiss.&

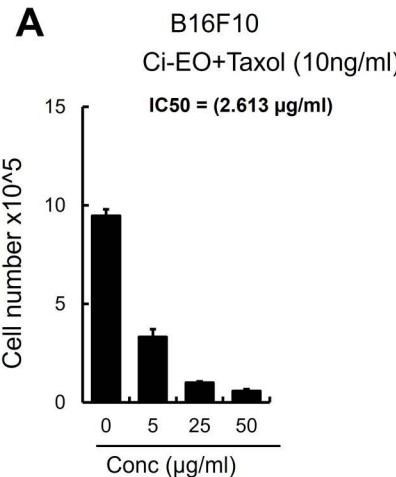
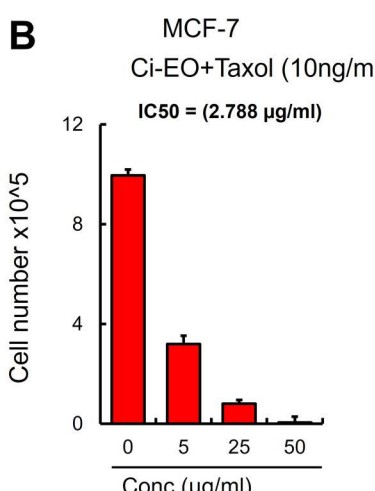

**Fig 5. (A. B) Viable B16F10 and MCF-7 cells were counted using Trypan blue after 24 h treatment of EO's and Taxol (10ng/ml) (n = 6/group).** Data are expressed as mean +/- SEM. **p 0.01; *p 0.05 (Student's t-test).

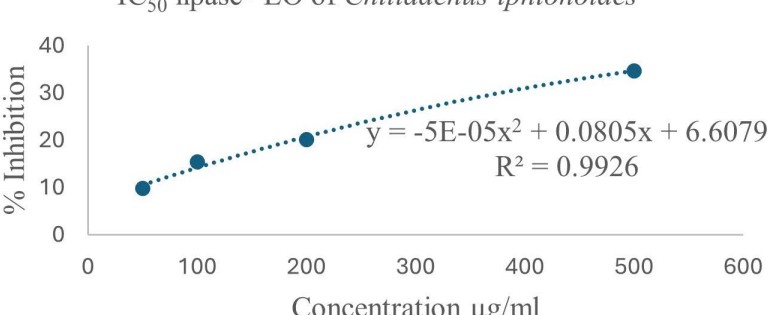

**Fig 6. A plot of percent lipase inhibition against the concentrations of essential oil extracted from dried *Chiliadenus iphionoides* (Boiss.& Blanche) Brullo leaves.**

Blanche) Brullo EO. The EO demonstrated a poor inhibition of α-amylase comparing to lipase inhibition. EO from dry leaves exhibited lower α-amylase inhibition, with an $IC_{50}$ of 1555.56 ± 77.77 µg/mL (Fig 7). The literature review indicated that the anti-lipase and anti-α-amylase properties of *Chiliadenus iphionoides* (Boiss.& Blanche) Brullo EO have not been previously explored, with only a single study addressing the α-amylase activities of its polar extract. The report indicates that the ethanol and water extracts of *Chiliadenus iphionoides* (Boiss.& Blanche) Brullo exhibited α-amylase inhibition rates of 70.50 ± 7.8% and 67.60 ± 7.0%, respectively, at a concentration of 200 mg/ml. The ethanol and water extracts' high flavonoid levels, which are absent in the essential oil, explain their effective α-amylase inhibition.

## 5. Conclusion

In this study, the essential oils extracted from the hydrodistillation of dry *Chiliadenus iphionoides* (Boiss.& Blanche) Brullo leaves were analyzed using GC-MS; 47 compounds were identified, accounting for 98.81% of the total oil, with cresol methyl ether (52.93%), ethyl 2-octynoate (14.36%), epi-α-cadinol (6.56%), 1,8-cineole (4.25%), and 7-epi-a-eudesmol (3.66%). The biological results demonstrated high antioxidant activity of the EO against 1,1-diphenyl-2-picrylhydrzyl (DPPH) and remarkable

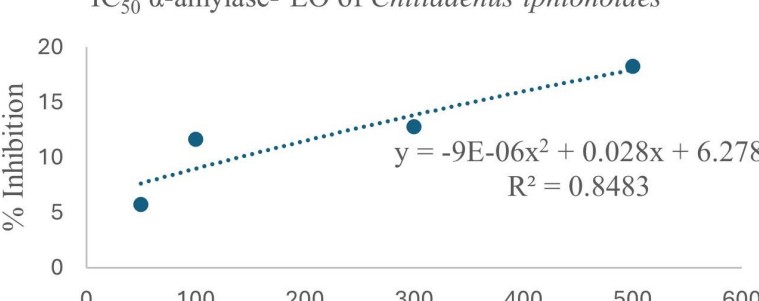

IC$_{50}$ α-amylase- EO of *Chiliadenus iphionoides*

$$y = -9E\text{-}06x^2 + 0.028x + 6.278$$
$$R^2 = 0.8483$$

**Fig 7. A plot of percent α-amylase inhibition against the concentrations of essential oil extracted from dried *Chiliadenus iphionoides* (Boiss.& Blanche) Brullo leaves.**

anticancer properties against MCF-7 and B16F10 cell lines. A 1:1 mixture of *Chiliadenus iphionoides* (Boiss.& Blanche) Brullo and *Teucrium polium L.* EOs improved their cytotoxicity, and the addition of the *Chiliadenus iphionoides* (Boiss.& Blanche) Brullo EO to Taxol caused a significant increase in cancer cell inhibition. However, it showed modest lipase activity and poor α-amylase activity. Our findings demonstrate that essential oils significantly inhibit the proliferation of both B16F10 and MCF-7 cell lines. Furthermore, the results suggest that combining EO with chemotherapeutic agents such as Taxol may offer a novel and potentially more effective therapeutic strategy against cancer. Finally, conducting in vivo and preclinical investigations will be essential to validate our results and comprehend *Chiliadenus iphionoides* (Boiss.& Blanche) Brullo EO mechanisms of action and potential therapeutic applications. This research may substantially advance the field of cancer treatment.

## Acknowledgments

We appreciate An-Najah National University.

## Author contributions

**Conceptualization:** Odey Bsharat, Nawaf Al-Maharik.

**Data curation:** Odey Bsharat, Nawaf Al-Maharik.

**Formal analysis:** Odey Bsharat, Nawaf Al-Maharik.

**Investigation:** Odey Bsharat, Nawaf Al-Maharik.

**Methodology:** Odey Bsharat, Nawaf Al-Maharik.

**Project administration:** Odey Bsharat, Nawaf Al-Maharik.

**Resources:** Odey Bsharat, Nawaf Al-Maharik.

**Software:** Odey Bsharat, Yousef Salama, Nawaf Al-Maharik.

**Supervision:** Odey Bsharat, Nawaf Al-Maharik.

**Validation:** Odey Bsharat, Yousef Salama, Nisreen Al Hajj, Nawaf Al-Maharik.

**Visualization:** Odey Bsharat, Yousef Salama, Nawaf Al-Maharik.

**Writing – original draft:** Odey Bsharat, Nawaf Al-Maharik.

**Writing – review & editing:** Odey Bsharat, Nisreen Al Hajj, Nawaf Al-Maharik.

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
