## [Decision Letter · Decision Letter 0]

18 May 2025

PONE-D-25-21407Chiliadenus iphionoides: From Chemical Profiling to Anticancer, Antioxidant, α-amylase, and α-glycosidase activitiesPLOS ONE

Dear Dr. Bsharat,

Thank you for submitting your manuscript to PLOS ONE. After careful consideration, we feel that it has merit but does not fully meet PLOS ONE’s publication criteria as it currently stands. Therefore, we invite you to submit a revised version of the manuscript that addresses the points raised during the review process.

We look forward to receiving your revised manuscript.

Kind regards,

Uğur Cakilcioğlu, PhD

Academic Editor

PLOS ONE

Reviewers' comments:

Reviewer's Responses to Questions

**Comments to the Author**

1. Is the manuscript technically sound, and do the data support the conclusions?

Reviewer #1: Yes

Reviewer #2: Yes

2. Has the statistical analysis been performed appropriately and rigorously? 

Reviewer #1: Yes

Reviewer #2: Yes

3. Have the authors made all data underlying the findings in their manuscript fully available?

Reviewer #1: Yes

Reviewer #2: Yes

4. Is the manuscript presented in an intelligible fashion and written in standard English?

Reviewer #1: Yes

Reviewer #2: Yes

5. Review Comments to the Author

Reviewer #1: There are many studies on the antioxidative and anticancer effects of Chiliadenus iphionoides extracts. In this manuscript, the essential oils were isolated and their chemical content was determined, and the antimicrobial, antioxidative and anticancer effects of this oil were investigated. In this respect, the originality of the study is high. The study was well planned and appropriate methods were used. Plant names should be written in italics in the References section. In Fig. 3 and 4, A and B should be separate or B should be rearranged. Fig. 5 is not of good quality, it should be redrawn and rearranged. In Figs. 6 and 7, the fonts of the x and y axes should be the same.

Reviewer #2: Dear Editor; The attached article was checked. The manuscript contains interesting information about Chiliadenus iphionoides: From Chemical Profiling to Anticancer, Antioxidant, α-amylase, and α-glycosidase activities

I think that this article is well suited to your journal.

It is generally good work. The scientific and presentation level of the manuscript is high.

The title is understandable and in line with the text. The text is written in a descriptive and understandable language. The material and method are well described and adequately detailed Discussion and conclusion are interrelated.

What is the difference between the study and existing studies?

-add to the introduction; available in different on the wild edible plant studies

https://doi.org/10.1002/ardp.202300263

https://doi.org/10.1002/ardp.202300528

https://doi.org/10.1002/ardp.202400194

https://doi.org/10.1002/efd2.70021

In ms: Write the name of the authority to the end of the plant’s name.

- http://www.theplantlist.org/

Please, read the paper and correct them all.

What is the difference between the study and existing studies?

References were cross-checked.

-The paper should be edited according to the writing rules of the journal

Original manuscript. There are, however, a few minor changes required.

6. PLOS authors have the option to publish the peer review history of their article (what does this mean? ). If published, this will include your full peer review and any attached files.

**Do you want your identity to be public for this peer review?** For information about this choice, including consent withdrawal, please see our Privacy Policy .

Reviewer #1: No

Reviewer #2: No

---

## [Author Response · Author response to Decision Letter 1]

23 May 2025

Thank you for you valuable Suggestions and comments.

Response to Editor:

Response: Thank you for your suggestion. Done according to Journal guidelines sections are divided into Heading 1, heading 2 and heading 3 as requested and, Title, References, affiliation all have been edited according to Journal guidelines.

Response: Done

Response to Reviewers:

Reviewer #1: There are many studies on the antioxidative and anticancer effects of Chiliadenus iphionoides extracts. In this manuscript, the essential oils were isolated and their chemical content was determined, and the antimicrobial, antioxidative and anticancer effects of this oil were investigated. In this respect, the originality of the study is high.

Thank you for your comments and suggestions.

1. The study was well planned and appropriate methods were used. Plant names should be written in italics in the References section.

Response: Done

2. In Fig. 3 and 4, A and B should be separate or B should be rearranged.

Response: Done, in Fig. 3 and 4, A and B have been separated.

3. Fig. 5 is not of good quality; it should be redrawn and

Response: Done Fig. 5 we redraw and rearrange it.

4. In Figs. 6 and 7, the fonts of the x and y axes should be the same.

Response: Thanks for catching this. Done, both of them are the same font and size.

Reviewer #2: Dear Editor; The attached article was checked. The manuscript contains interesting information about Chiliadenus iphionoides: From Chemical Profiling to Anticancer, Antioxidant, α-amylase, and α-glycosidase activities. I think that this article is well suited to your journal. It is generally good work. The scientific and presentation level of the manuscript is high. The title is understandable and in line with the text. The text is written in a descriptive and understandable language. The material and method are well described and adequately detailed Discussion and conclusion are interrelated.

Thank you for your comments and suggestions.

1. What is the difference between the study and existing studies?

Response:

-This is an excellent question. The plants studied were collected from Jericho, the lowest place on Earth and located very close to the Dead Sea. The soil in Jericho is unique, containing higher salt concentrations compared to other regions. Additionally, the annual precipitation in Jericho is approximately 200 ml. Based on our previous experience, the chemical composition of plants from Jericho differs from that of essential oils collected from other locations.

-Most prior studies have focused on the chemical composition of C. iphionoides essential oil and its antioxidant activity, but only two studies have addressed its anticancer activity. Furthermore, there are currently no studies investigating its α-glucosidase and lipase inhibitory activities. This gap in research presents an opportunity to explore the potential therapeutic benefits of C. iphionoides essential oil. Investigating its effects on α-glucosidase and lipase could reveal valuable insights into its role in managing metabolic disorders and cancer treatment.

- Also, we add a paragraph on the main manuscript line 226-230

2. add to the introduction, available in different on the wild edible plant studies

Response: we added these references as requested. Now they are Ref 4, 5 and 14 in the main and highlighted manuscript.

https://doi.org/10.1002/ardp.202300263

https://doi.org/10.1002/ardp.202400194

https://doi.org/10.1002/efd2.70021

Thank you for the suggested references. Done as requested

3. In ms: Write the name of the authority to the end of the plant’s name.

- http://www.theplantlist.org/

Please, read the paper and correct them all.

Response: Done as requested

4. The paper should be edited according to the writing rules of the journal

Original manuscript. There are, however, a few minor changes required.

Response: Done as requested

---

## [Decision Letter · Decision Letter 1]

19 June 2025

Chiliadenus iphionoides: From Chemical Profiling to Anticancer, Antioxidant, α-amylase, and α-glycosidase activities

PONE-D-25-21407R1

Dear Dr. Bsharat,

We’re pleased to inform you that your manuscript has been judged scientifically suitable for publication and will be formally accepted for publication once it meets all outstanding technical requirements.

Kind regards,

Uğur Cakilcioğlu, PhD

Academic Editor

PLOS ONE

Additional Editor Comments (optional):

Reviewers' comments:

Reviewer's Responses to Questions

**Comments to the Author**

1. If the authors have adequately addressed your comments raised in a previous round of review and you feel that this manuscript is now acceptable for publication, you may indicate that here to bypass the “Comments to the Author” section, enter your conflict of interest statement in the “Confidential to Editor” section, and submit your "Accept" recommendation.

Reviewer #1: All comments have been addressed

Reviewer #2: (No Response)

2. Is the manuscript technically sound, and do the data support the conclusions?

Reviewer #1: Yes

Reviewer #2: (No Response)

3. Has the statistical analysis been performed appropriately and rigorously? 

Reviewer #1: Yes

Reviewer #2: Yes

4. Have the authors made all data underlying the findings in their manuscript fully available?

Reviewer #1: Yes

Reviewer #2: Yes

5. Is the manuscript presented in an intelligible fashion and written in standard English?

Reviewer #1: Yes

Reviewer #2: Yes

6. Review Comments to the Author

Reviewer #1: Although there are many studies on the metabolite content and antioxidative antimicrobial effects of the extracts of various organs of Chiliadenus iphionoides in the manuscript, there is not enough research on the antioxidative, antimicrobial effects as well as anticancer effects of its essential oils. This situation increases the original value of the manuscript.

The authors have made the requested corrections. It is suitable for publication.

Reviewer #2: (No Response)

7. PLOS authors have the option to publish the peer review history of their article (what does this mean? ). If published, this will include your full peer review and any attached files.

**Do you want your identity to be public for this peer review?** For information about this choice, including consent withdrawal, please see our Privacy Policy .

Reviewer #1: **Yes: ** YASEMİN ÖZDENER KÖMPE

Reviewer #2: No

---

## [Editor Report · Acceptance letter]

PONE-D-25-21407R1

PLOS ONE

Dear Dr. Bsharat,

I'm pleased to inform you that your manuscript has been deemed suitable for publication in PLOS ONE. Congratulations! Your manuscript is now being handed over to our production team.

Kind regards,

on behalf of

Professor Uğur Cakilcioğlu

Academic Editor

PLOS ONE